# Adsorbent Precoating by Lyophilization: A Novel Green Solvent Technique to Enhance Cinnarizine Release from Solid Self-Nanoemulsifying Drug Delivery Systems (S-SNEDDS)

**DOI:** 10.3390/pharmaceutics15010134

**Published:** 2022-12-30

**Authors:** Ahmad Yousef Tashish, Ahmad Abdul-Wahhab Shahba, Fars Kaed Alanazi, Mohsin Kazi

**Affiliations:** 1Department of Pharmaceutics, College of Pharmacy, King Saud University, P.O. Box 2457, Riyadh 11451, Saudi Arabia; 2Kayyali Research Chair for Pharmaceutical Industries, College of Pharmacy, King Saud University, Riyadh 11451, Saudi Arabia

**Keywords:** adsorbent curing, S-SNEDDS, solid self-nanoemuslifying drug delivery systems, cinnarizine, solidification

## Abstract

Background: Solidification by high surface area adsorbents has been associated with major obstacles in drug release. Accordingly, new approaches are highly demanded to solve these limitations. The current study proposes to improve the drug release of solidified self-nanoemulsifying drug delivery systems (SNEDDS) to present dual enhancement of drug solubilization and formulation stabilization, using cinnarizine (CN) as a model drug. Methods: The solidification process involved the precoating of adsorbent by lyophilization of the aqueous dispersion of polymer–adsorbent mixture using water as a green solvent. Then, the precoated adsorbent was mixed with drug-loaded liquid SNEDDS to prepare solid SNEDDS. The solid-state characterization of developed cured S-SNEDDS was done using X-ray powder diffraction (XRD) and differential scanning calorimetry (DSC). In vitro dissolution studies were conducted to investigate CN SNEDDS performance at pH 1.2 and 6.8. The solidified formulations were characterized by Brunauer–Emmett–Teller (BET), powder flow properties, scanning electron microscopy, and droplet size analysis. In addition, the optimized formulations were evaluated through in vitro lipolysis and stability studies. Results: The cured solid SNEDDS formula by PVP k30 showed acceptable self-emulsification and powder flow properties. XRD and DSC revealed that CN was successfully amorphized into drug-loaded S-SNEDDS. The uncured solid SNEDDS experienced negligible drug release (only 5% drug release after 2 h), while the cured S-SNEDDS showed up to 12-fold enhancement of total drug release (at 2 h) compared to the uncured counterpart. However, the cured S- SNEDDS showed considerable CN degradation and decrease in drug release upon storage in accelerated conditions. Conclusions: The implemented solidification approach offers a promising technique to minimize the adverse effect of adsorbent on drug release and accomplish improved drug release from solidified SNEDDS.

## 1. Introduction

Self-nanoemulsifying drug delivery systems (SNEDDS) have been a buzzword to express ease of preparation and unmatched nanoformulation features [1]. Liquid SNEDDS introduce a vital technique for formulating poorly-water soluble drugs (PWSD) because the former provides substantial enhancement of their dissolution and bioavailability and is less affected by pH change [2]. Nevertheless, liquid SNEDDS (L-SNEDDS) are associated with some limitations such as the possibility of leakage from the capsule, capsule shell incompatibilities with some liquid excipients, oil rancidity, and drug precipitation [3]. In addition, certain drugs might chemically degrade when exposed to SNEDDS excipients [4,5], whereas Solid SNEDDS (S-SNEDDS) could preserve the solubilization benefits of SNEDDS along with less formulation limitations, better stability, and patient compliance [1,6]. Among various SNEDDS solidification techniques, adsorption onto inorganic silica materials offers the advantages of simplicity, ease of preparation, and produce free-flowing powders within a few seconds. Nevertheless, this technique has been associated with substantial hindrance of drug release from the solid formulation. Many studies have reported the adverse effect of adsorbent on the release of the drug from solid SNEDDS [7,8,9].

Different mechanisms could be responsible for the observed drug release retardation upon using adsorbents for SNEDDS solidification. The drug release retardation could be attributed to SNEDDS blockade within the meso pores of the adsorbent either due to SNEDDS-induced gel formation or the very small (2–50 nm) pore size of the mesoporous region which impedes the emulsification process compared to the macroporous region (this allows more space for emulsification process due to its larger pore size (>50 nm) of macropores). Furthermore, some studies suggested that the developed physical bonds between the drug and the carrier could cause drug diffusion to the adsorbent surface leading to drug precipitation out of the SNEDDS droplets and therefore incomplete drug release from the S-SNEDDS [10,11,12].

Therefore, it is imperative to adopt new approaches that are capable of enhancing drug release from S-SNEDDS. In this context, the current study aimed to explore a new approach that involves adsorbent precoating with hydrophilic polymer using lyophilization of the aqueous dispersion of a polymer–adsorbent mixture. The technique was considered environmentally green due to the circumvent of organic solvent in preparation. Then, the precoated adsorbent was mixed with drug-loaded L-SNEDDS to achieve S-SNEDDS (Figure 1). This approach is hypothesized to relatively block the small pores in the mesoporous regions by applying a hydrophilic polymer/precipitation inhibitor. Accordingly, SNEDDS might not penetrate and/or be trapped into mesoporous regions and, therefore, drug–adsorbent interaction could be minimized leading to favorable drug release enhancement. In addition, the hydrophobic surface of the adsorbent could be substantially masked by a hydrophilic polymer and/or precipitation inhibitor. Consequently, drug precipitation could be reduced by inhibiting the unfavorable interaction between the drug and silica. Another mechanism of using hydrophilic polymer could be the increase of water penetration into the silicate by wicking action and, thereby, facilitating drug release. Cinnarizine (CN), a weakly basic PWSD, is limited by poor/pH-dependent aqueous solubility and chemical instability in lipid-based excipients [4,5]. Thus, CN is expected to show limited dissolution within hypochlorhydria/achlorhydria conditions (higher pH environments) as proved in the previous studies [2,13]. In the current work, cured S-SNEDDS was developed using cured adsorbents and in vitro dissolution studies were conducted to evaluate the impact of different curation parameters on drug release compared to the conventional uncured S-SNEDDS. Within the scope of the current study, cured CN-SNEDDS were evaluated at a high pH dissolution environment (pH 6.8) to mimic an extreme achlorhydria condition [14,15]. 

## 2. Materials and Methods

### 2.1. Materials

Kolliphor EL (K-EL), Soluplus, and polyvinylpyrrolidone (PVP: Kollidon K30 and K90) were obtained from BASF (Ludwigshafen, Germany). Imwitor 308 (I308) was generously donated by Sasol Germany GmbH (Werk, Witten, Germany). Avochem (Cheshire, UK) supplied the oleic acid (OL). FDC Limited (Maharashtra, India) provided cinnarizine (CN, purity > 99.5%). Capsugel (Morristown, NJ, USA) generously provided fish gelatin size 0 capsules. Grace (Worms, Germany) provided Syloid^®^ SP 53D-11786 (SYL, amorphous magnesium alumino-metasilicate). JRS Pharma (Rosenberg, Germany) supplied the Vivapharm^®^ Hydroxy-propyl methyl cellulose (HPMC) E3. Neusilin^®^ grades US2 and UFL2 were donated by Fuji Chemical Industries (Osaka, Japan).

### 2.2. Drug-Free and Drug-Loaded Liquid Self-Nanoemulsifying Drug Delivery Systems (L-SNEDDS) Preparation

L-SNEDDS were prepared using the excipients OL/I308/K-El (at %25/25/50, *w*/*w*/*w* ratios) (Table 1). The cosurfactant (I308) was first preheated to ensure complete melting and homogenization before being mixed with the oil and surfactant components. For 10 min, the mixture was stirred at 1250 rpm, then the drug was dissolved in the formulation at a concentration of ≈ 80 mg/g (Table 1) and blended as previously described described [16,17,18]. 

### 2.3. Precoating (Curing) of Adsorbent Using Hydrophilic Polymers

Several hydrophilic polymers were screened including PVP-K30, PVP-K90, HPMC E3, and soluplus. The selected polymer was weighed (800 mg) and dissolved in 50 mL of aqueous solution at different pH. Then, 3.2 g of the adsorbent was introduced to the aqueous solution of the polymer and blended to achieve a slurry-like consistency. At that point, the resultant dispersion was lyophilized for at least 48 h at −60 °C (Alpha 1-4 LD Plus, Osterode am Harz, Germany). To create a fine powder with uniformed size, the lyophilized powder (cured adsorbent) was ground by hand using a mortar and pestle and allowed to pass through a 315 m sieve [2].

### 2.4. Preparation of Cured Solid Self-Nanoemulsifying Drug Delivery Systems 

L-SNEDDS was mixed with a predetermined quantity of the cured adsorbent Syloid in a 1:1 (*w/w*) ratio. A uniform powder was then produced by thoroughly blending the mixture for 10 min at 700 rpm (L32 Labinco magnetic stirrer, Labinco B.V., Breda, The Netherlands) [19]. Following that, the cured solidified SNEDDSs were characterized in order to attain the optimal formulation.

### 2.5. Determination of CN Encapsulation Efficiency

A preset quantity of CN-loaded S-SNEDDS was weighed and transferred into a volumetric flask (25 mL capacity), which was then filled with acetonitrile. Complete drug solubilization in the solvent was confirmed by sonication up to 45 min. Subsequently, the solution was transferred into a 1.5 mL Eppendorf tube and centrifuged at 10,000 rpm for 5 min. At that point, an aliquot of the supernatant was withdrawn and analyzed by UPLC [20]. Each sample was analyzed in triplicates. 

### 2.6. Optimization and Characterization of S-SNEDDS

#### 2.6.1. Powder Properties

Using a tapped density tester-USP (Erweka SVM 102, Erweka GmbH, Heusenstamm, Germnay) to measure the bulk and tapped densities of S-SNEDDS, 0.6 g of sample was poured into a graduated cylinder with 0.2 mL markings that had a ten-mL capacity. Additionally, the Hausner ratio and compressibility index were calculated using previously published methods [21,22,23]. In addition, the height funnel method was used to determine the angle of repose [22,23].

#### 2.6.2. Differential Scanning Calorimetry (DSC)

Cured SNEDDS with different polymer ratios were analyzed using a differential scanning calorimeter (DSC8000, Elmer, Waltham, MA, USA). Samples (3-5 mg) were collected and hermetically sealed in aluminum pans using a crimp sealer. Heat was then applied to the sealed pans between 25 and 200 °C, at a heating rate = 10 °C/min and under nitrogen gas (flow rate = 50 mL/min) [10].

#### 2.6.3. X-ray Powder Diffraction (XRD)

The samples for XRD analysis were evaluated by a diffractometer (Model: Ultima IV, Rigaku Corporation, Tokyo, Japan) over 3–30° 2θ range at 0.5 deg./min scan speed. The tube anode was Cu with Ka = 0.154 nm monochromatized with a graphite crystal. The pattern was collected at a 40 kV tube voltage and a 40 mA tube current in step scan mode (counting time = 1 s/step and step size = 0.02°) [18].

#### 2.6.4. Analysis of Droplet Size, PDI, and Zeta Potential of L-SNEDDS and S-SNEDDS

Cured S-SNEDDS samples were dispersed in distilled water at a 1:1000 *w*/*w* mixture ratio, followed by 5 min stirring at 1000 rpm to ensure complete formulation distribution [18]. Preceding the analysis, the formed aqueous suspension was centrifuged to separate any undissolved particles (arising from solid excipients) that might interfere with the droplet size. Finally, the mean droplet size, polydispersity index (PDI), and zeta potential (ZP) value of diluted SNEDDS formulations were measured using a Zetasizer Nano ZS (Malvern Panalytical Ltd, Malvern, Worcestershire, UK). The data were represented as the mean of three replicates [24].

#### 2.6.5. Scanning Electron Microscopy (SEM)

The solid powder samples (pure CN, pure SYL, cured SYL [before SNEDDS loading]), and cured S-SNEDDS) were examined using a scanning electron microscope (Carl Zeiss EVO LS10, Cambridge, UK). S-SNEDDS were assessed in terms of morphological characteristics of the formulation and to detect signs of poor solidification attributes. Samples were coated with gold in a Q150R sputter coater unit (Quorum Technologies Ltd., Lewes, UK) under vacuum for 60 sec under 20 mA argon atmosphere [19,25].

#### 2.6.6. The Brunauer–Emmett–Teller (BET)

The BET method was adopted to calculate the specific surface area of S-SNEDDS using adsorption data at a relative pressure from 0.02 to 0.20. By using the Barrett–Joyner–Halenda (BJH) model, pores volume and size distributions were measured from the adsorption branches of isotherms. Total pores volume was estimated from the adsorbed amount of nitrogen molecules at a relative pressure P/P0 of 0.99514 [11].

### 2.7. In Vitro Dissolution Tests

The dissolution tests were conducted using an automated USP Type II dissolution apparatus (UDT-814, LOGAN Inst. Corp., Franklin, NJ, USA) at a paddle speed = 50 rpm and 37 °C temperature. The dissolution medium involved either 500 mL of phosphate buffer at pH 6.8 or 500 mL of 0.1M HCL (pH 1.2) with no enzymes. According to the European Pharmacopoeia 7.0, 0.1% *w*/*v* potassium dihydrogen phosphate, 0.2% *w*/*v* dipotassium hydrogen phosphate, and 0.85% *w/v* sodium chloride were used to make the phosphate buffer (pH 6.8). In the current dissolution study, CN/PVP-K30 physical mixture (≈1:2.5 *w*/*w*) was utilized as a control, and various uncured and cured SNEDDS (≈25 mg CN equivalent) were examined to evaluate the influence of different curing parameters on CN release. In addition, A minimum of three replicates were tested for each formulation, and serial samples (2 mL) were taken and centrifuged at intervals of 5, 10, 15, 30, 60, and 120 min. An aliquot of the supernatant was properly diluted with acetonitrile and subjected to UPLC analysis [2].

### 2.8. In Vitro Lipolysis

For each lipolysis test, 500 mg of CN-loaded liquid formulation (or an equivalent amount of cured S-SNEDDS) was dispersed into 18 mL of a digestion buffer under fed state conditions (101 mM NaOH, 144 mM Glacial acetic acid, 203 mM NaCl, pH 5.0). SIF powder containing bile salt (taurocholate) and phospholipid (lecithin) was added at 4:1 (ratio secreted in bile) molar ratio in the digestion mixture [26]. The representative SNEDDSs were emulsified in the mixed micellar solutions prior to enzyme addition in the thermostatic jacketed glass reaction vessel. Lipolysis was initiated after the addition of 2 mL of pancreatin extract in the reaction vessel [27]. Lipolysis was performed at 37 °C and allowed to continue for 30 min at constant pH of 6.8 using a pH-stat titration unit (902 Titrando, Metrohm AG, Herisau, Switzerland). The fatty acids produced during the lipolysis reaction were titrated with 0.2 M NaOH for all the formulations.

At 0 time, 5 min, 15 min, and 30 min intervals, 1 mL sample solution was withdrawn and centrifuged for 5 min at 15,000 rpm. Subsequently, a 100 µL aliquot was taken from the supernatant, diluted into 900 µL of acetonitrile, and then analyzed by UPLC as described previously [20]. Each sample was analyzed in triplicates.

#### Initial Digestion Rate Evaluation

The slopes of the hydrolysis curve at the beginning of the reaction were determined by linear regression analysis in order to calculate the initial reaction rate. Up until the slope value of these straight lines started to decrease, a number of experimental points were included, and the slope was calculated using the least-squares linear regression method [27]. Following the addition of the pancreatin solution into the reaction vessel, the most stable estimate of the slope was typically seen 0–3 min later.

### 2.9. Accelerated Stability Studies

The cured S-SNEDDS (with 2:8 and 3:7 ratios of polymer to adsorbent) were enrolled in the stability studies to evaluate their performance upon storage in accelerated conditions. The formulations were stored in a stability cabinet (Binder Gmbh, Tuttlingen, Germany). The storage conditions were maintained, according to the ICH guidelines, with relative humidity (RH) of 75 ± 5) at 40 °C [4,5]. Samples were taken out at predetermined intervals (0, and 3 months) and allowed to reach room temperature. Withdrawn samples were then diluted in acetonitrile and finally assayed by UPLC to determine the cinnarizine concentration. The degradation of cinnarizine was evaluated according to the changes in intact drug concentrations. Three replicates were considered for each sample. In addition, the formulation physical appearance was examined to record any signs of agglomeration or color change. Most importantly, dissolution and XRD studies were also conducted to evaluate the change in dissolution profile and drug crystallinity upon storage.

### 2.10. CN Quantification by UPLC Assay

With a minor modification, CN was quantified using a validated reversed-phase UPLC method [20]. To achieve good separation between the CN and degradation product peaks, the method was adjusted to have a mobile phase composition of 0.5% trifluoracetic acid: acetonitrile (55:45). Peak separation was accomplished using an Acquity^®^ UPLC BEH C18 (2.1 50 mm, 1.7 m) column coupled to an acquity guard filter. The flow rate was maintained at 0.25 mL/min and the UV detector was set at 251 nm.

### 2.11. Statistical Analysis

SPSS 26 software was in use to check for the significance of the data. A one-way ANOVA followed by post hoc test “LSD” were used to compare droplet size and zeta potential parameters. A value of *p* < 0.05 was denoted as significant throughout the study [13].

## 3. Results

### 3.1. Characterization of Solid SNEDDS

#### 3.1.1. Powder Properties

Pure SYL showed fair to excellent flow properties among different powder flow properties (Table 2). Both cured adsorbent and cured S-SNEDDS showed acceptable flow properties with no signs of particle agglomeration at 10% and 20% PVP ratios. However, higher PVP ratios (30–40%) showed serious adverse on the adsorbent powder flow properties, particularly before SNEDDS addition, as represented by cured SYL (10–40% PVP) (Table 2). The resulting S-SNEDDS after L-SNEDDS adsorption on cured SYL fully solidified as free-flowing powder and displayed acceptable flow properties that ranged from passable to good, as per the USP guidelines (Table 2) [28].

#### 3.1.2. Differential Scanning Calorimetry (DSC)

The results from the DSC analysis confirmed that pure CN showed a sharp endothermic peak at 125 °C (Figure 2), confirming its crystallinity [29], while pure SYL and all the cured formulas showed complete disappearance of the CN peak at the same temperature range.

#### 3.1.3. X-ray Diffraction (XRD)

The XRD results represented in Figure 3 showed typical X-ray diffraction peaks for pure CN at 3° to 30° (2θ). Similar to DSC, it also confirmed the crystalline state of CN. Both pure Syloid and cured S-SNEDDS (different ratios) showed a complete absence of sharp CN diffraction peaks.

#### 3.1.4. Droplet Size and Zeta Potential

Uncured S-SNEDDS showed significantly higher droplet sizes compared to liquid SNEDDS and 20% and 30% cured S-SNEDDS formulations (Figure 4). Interestingly, the L-SNEDDS exhibited a significant reduction in both formulation droplet size and PDI, compared to the uncured and cured S-SNEDDS counterparts (Figure 4 and Figure 5).

On the other hand, solidified S-SNEDDS showed a significant increase in the ZP value compared to liquid SNEDDS as follows: drug-loaded L-SNEDDS < uncured S-SNEDDS < 20% cured S-SNEDDS < 30% cured S-SNEDDS with significant differences between each formulation (Figure 6).

#### 3.1.5. Scanning Electron Microscopy (SEM)

Pure CN and pure SYL showed discrete irregular particles (Figure 7A,B). Except for cured SYL (30% PVP), no significant changes in particle morphology were observed after curing raw adsorbent nor mixing with SNEDDS (Figure 7C–H). The particles remained discrete, with no signs of incomplete solidification or agglomeration.

#### 3.1.6. The Brunauer–Emmett–Teller (BET) Analysis

All cured adsorbent samples showed a considerable drop in total surface area, pore volume, and average pore size compared to pure SYL. In particular, the 10%, 20%, and 30% cured SYL by PVP-K30 showed a gradual decrease in total surface area, pore volume, and pore size upon increasing the PVP polymer ratio from 10 to 30% (Table 3).

### 3.2. In Vitro Dissolution

#### 3.2.1. Effect of the Adsorbent Type on CN Release

Drug-loaded L-SNEDDS showed enhanced CN release (up to 69%) at pH 6.8. In contrast, the uncured S-SNEDDS comprising the adsorbent SYL showed poor CN release (<13%) up to 2 h at pH 6.8 (Figure 8A), while the uncured S-SNEDDS comprising NUS-US2 and NUS-UFL2 grades showed enhanced CN release up to 29 and 39%, respectively. However, the formulation comprising NUS-UFL2 showed a wet mass and incomplete solidification upon mixing with L-SNEDDS at a 1:1 ratio, and hence, it was excluded from the subsequent optimization steps within the current study.

#### 3.2.2. Effect of PVP Physical Mixture and Different Adsorbents Curing on CN Release

PVP physical mixture showed negligible CN release (up 3%) within 2 h. Both uncured S-SNEDDS (comprising SYL and NUS-US2 showed superior (up to 13% and 29%) CN release compared to the PVP physical mixture (Figure 8B,C). Surprisingly, the cured S-SNEDDS comprising cured SYL showed enhanced CN release (up to 57%) compared to cured S-SNEDDS comprising NUS-US2 (up to 40%) (Figure 8B,C). In fact, the cured S-SNEDDS (comprising SYL) showed a more than four-fold increase in CN release compared to its uncured SYL counterpart. On the other hand, the cured S-SNEDDS (comprising NUS-US2) showed less than 38% increment in maximum CN release CN compared to its uncured NUS-US2 counterpart.

#### 3.2.3. Effect of Curing Polymer (PVP-k30) Ratio on CN Release

The cured S-SNEDDS (comprising SYL cured by 10% PVP-k30) showed a maximum of 47% CN release (Figure 8C), which was increased to 56% and 68% release in the case of using 20% and 30% PVP-K30. In contrast, the uncured SYL formula showed a negligible CN release (less than 13%) within 2 h.

### 3.3. In Vitro Lipolysis

In FeSSIF conditions, liquid SNEDDS showed complete CN solubilization at the initial stage of the reaction. Upon the addition of pancreatin, the lipolysis process started and the amount of solubilized CN in the aqueous phase was decreased to 88% by the end of the experiment (Figure 9). On the other hand, the cured S-SNEDDS formulation was able to release 59% at the initial reaction stage. Upon pancreatin addition, the lipolysis started, and the solubilized CN amount in the aqueous phase was almost similar (≈ 58%) by the study end.

The lipid digestion progress was generally monitored by quantifying the digestion rate and extent indirectly via titration of the fatty acid produced [27]. The digestion profiles illustrated the mole fraction of titrated NaOH and total available free fatty acid (FA) under fed conditions. For liquid SNEDDS and cured S-SNEDDS, the initial (0–10 min) rate of reaction was rapid and accounted for over 25% of the final mole fraction of titrated NaOH and total available fatty acid (FA) liberated during 30 min (Figure 10 and Figure 11). For drug-loaded liquid SNEDDS, the lipolysis rate dropped off steeply under fed conditions and become approximately linear after ~18 min (Figure 10A and Figure 11A). Instead, the initial rapid digestion rate (0–10 min) continued similarly over the 30 min period in the case of cured S-SNEDDS (Figure 10B and Figure 11B). The initial digestion rate of liquid SNEDDS and cured S-SNEDDS was almost similar in fed state conditions (Figure 10 and Figure 11).

### 3.4. Accelerated Stability Studies

The 20% cured S-SNEDDS maintained > 78 % of intact CN upon storage in accelerated conditions for 3 months. Similarly, the 30% cured S-SNEDDS formulation maintained more than 70% of intact CN up to 3 months (Figure 12). Furthermore, when comparing the dissolution profile of each formulation at the initial and 3 months, the two dissolution profiles were slightly decreased at pH 1.2 (Figure 13A). On the other hand, the CN dissolution was significantly diminished from the 3 months sample at pH 6.8 (Figure 13B). The initial sample of 20% cured S-SNEDDS showed up to 56% dissolution compared to the 3 months sample that showed a maximum of 22% CN release, up to 2 h (at pH 6.8) (Figure 13B). Similarly, the 30% cured S-SNEDDS formulation showed up to 68% dissolution compared to the 3 months sample that showed a maximum of 26% CN release for up to 2 h (at pH 6.8). Finally, the powder color changed from beige to yellow after 3 months of storage in accelerated conditions, while no other significant changes in physical appearance or agglomeration were observed (Figure 14).

## 4. Discussion

The adsorption of L-SNEDDS to inorganic silica adsorbents has been limited by the significant impediment of drug release from the solid formulation [2,8,30]. Therefore, a new approach to enhance drug release from S-SNEDDS is highly demanded. The current study aims to explore a new approach that involves adsorbent curation with hydrophilic polymer using lyophilization of the aqueous dispersion of polymer–adsorbent mixture (Figure 1).

In the current study, several adsorbents, polymers (for adsorbent precoating), and preparation media have been evaluated in terms of their influence on SNEDDS solidification efficiency, powder flowability, and CN release from the solid formulation. Generally, the curing process was conducted using a hydrophilic polymer mixed with the adsorbent in an aqueous solution then the latter was lyophilized to obtain a free-flowing cured adsorbent powder. Subsequently, the cured adsorbent was mixed with L-SNEDDS to produce solidified SNEDDS powder. The conversion of S-SNEDDS from the representative L-SNEDDS using the cured adsorbents showed acceptable flow properties (passable to good), as confirmed by the SEM and powder flow properties findings, respectively.

The data from the DSC and XRD analyses showed successful CN amorphization within cured S-SNEDDS as no CN peaks were observed from the chromatograms. These findings confirm that CN was not exhibited as crystalline form, within solid SNEDDS, which ensures that CN was not precipitated during the solidification process from the L-SNEDDS formulation [17].

The higher droplet size upon solidification of SNEDDS could be due to the addition of insoluble inorganic silica carrier (SYL), which might interfere with nanoemulsion droplets upon aqueous dispersion and facilitate droplet size changes. On the other hand, cured S-SNEDDS contributed in a positive way to reducing droplet sizes during the solidification process. In fact, both cured solid SNEDDS (20% and 30% PVP-k30) showed a significant reduction in the droplet size of S-SNEDDS. 

The SEM data (Figure 7) reveals that the adsorbent SYL and cured S-SNEDDS showed discrete irregular particles that ranged in size from a few micrometers up to ≈245 µm. Upon exposure to GIT aqueous media, these microparticles are expected to undergo a rapid self-nanoemulsification process resulting in a fine nanoemulsion with an average size of ≈101 nm, as evidenced by the formulation droplet size findings (Figure 4). This in-situ transition from the microscale to nanoscale is valuable in terms of dissolution and absorption perspectives. This ultra-low nano droplet size could be linked with increased drug release rate, increased surface area available for drug absorption, and formation of readily digestible oil droplets that can be incorporated into mixed micelles and pass the intestinal lumen [19,31]. Compared to other systems of similar size, Rutkowski et al. reported a technique to prepare hydrogel alginate capsules (microparticles) that can be adjusted in size from 10 µm to 2 mm [32]. In addition, Sukhorokov reported a method to prepare multi-layer films of polyelectrolytes adsorbed onto charged polystyrene latex particles. These nanoparticles ranged in size from 100–260 nm depending on the number of layers applied [33]. In fact, the aforementioned microparticles [32] and nanoparticles [33] are quite different drug delivery systems that have different pharmaceutical applications other than the currently studied S-SNEDDS. However, it is worth mentioning that these cured S-SNEDDS offer an exclusive advantage of in-situ transition from the micro to nanoscale droplets upon exposure to GIT aqueous environment. Accordingly, the current cured S-SNEDDS gather the advantage of microparticles in terms of better physical stability and less particle aggregation on storage along with the advantages of nanoparticles (that are formed in-situ upon the self-emulsification process) in terms of significantly higher surface area available for drug absorption.

Another factor to consider when assessing emulsification effectiveness is zeta potential (ZP). The significance of the ZP value may be linked to the stability of the nanoemulsion. The existence of non-ionic surfactants, the adsorption of anionic species to the droplet surfaces, or the presence of free fatty acid impurities in the surfactant could all be contributing factors to the observed negative zeta potential values of all the SNEDDS [19]. The significant decrease in ZP values in cured solid SNEDDS, compared to L-SNEDDS, could be due to the presence of SYL and/or PVP-k30 in aqueous formulation dispersion.

Since SNEDDS emulsify spontaneously upon exposure to the dispersion medium, it is very likely that SNEDDS adsorbed on SYL may not be able to escape the pores of the silicate and enter the dispersion medium before the emulsification process initiates. In other words, it is necessary that the emulsification process should start within the pores of the silicates [30].

Regarding the in vitro dissolution studies, the CN/PVP physical mixture showed negligible CN release at pH 6.8. Although PVP-K30 is an effective precipitation inhibitor that has been widely used to enhance drug dissolution, it failed to enhance CN dissolution (at pH 6.8), when used solely. This finding might be owing to the challenging physicochemical property of CN; being a weak base with very poor solubility at neutral and basic media. Similar findings were reported in previous studies where CN/PVP solid dispersion showed significant precipitation and limited dissolution at pH 6.8 [18]. However, SNEDDS technology showed significant enhancement of CN dissolution at the same pH. This might be attributed to SNEDDS ability to form a favorable microenvironment within nanoemulsion droplets that were able to maintain CN solubilized and protect it from exposure to the unfavorable neutral environment that is associated with dramatic drug precipitation and limited release. 

On the other hand, S-SNEDDS using uncured adsorbent showed very low CN release compared to L-SNEDDS. Several optimization studies were performed to overcome the limitation of adsorbent on CN release from S-SNEDDS. In particular, the cured S-SNEDDS (comprising SYL cured by PVP K30) showed more than 4-fold increase in CN release compared to its uncured SYL counterpart. 

In contrast to CN/PVP physical mixture, the proper utilization of PVP-K30 in adsorbent precoating led to significant CN release enhancement. This finding confirms the vital role of PVP in the adsorbent precoating, which helped in overcoming the adsorbent adverse effect on drug release from solidified SNEDDS. On the other hand, the sole presence of PVP in the formula failed to enhance CN release while PVP played a vital role in overcoming the significant retardation of drug release from SNEDDS that has been attributed to the adsorbent. Among different polymers tested for adsorbent precoating (curing), PVP-K30 showed the best results in terms of CN release (>50%). Interestingly, increasing the proportion of polymer PVP-K30 showed a further increment in CN dissolution up to 60% for 30% cured S-SNEDDS. These findings support the previous studies that showed a higher drug release from PVP-coated formulations compared to their counterparts that lack PVP coating [8]. These data could be also correlated with the current BET results that showed a corresponding gradual decrease in the surface area upon increasing the PVP-K30 ratio (Table 3). These findings suggest that the adsorbent hydrophobic surface was significantly masked by the hydrophilic polymer, which led to minimal interaction between the drug and the silica and, therefore, minimal drug precipitation. Additionally, the hydrophilic polymer might promote water uptake into the silicate through wicking action, which could promote drug release [8,30]. Former studies also suggested a third mechanism to explain the enhanced drug release from cured S-SNEDDS by blocking the adsorbent small pores upon curing the adsorbent with hydrophilic polymers such as PVP-K30. Accordingly, SNEDDS might not penetrate and/or be trapped in mesoporous regions allowing less drug–adsorbent interaction and higher drug release. However, the current BET findings showed a gradual pore size decrease upon increasing the % of curing polymer PVP from 0 to 30%, which diminishes the leverage of the third mechanism to the enhanced drug release from cured S-SNEDDS [8,30].

Although the high PVP-K30 ratio in cured S-SNEDDS (30%) showed the highest CN release enhancement, this was associated with a negative impact on the flow properties of S-SNEDDS powder (Table 2). On the other hand, the 20% cured S- SNEDDS showed passable-good flow properties along with acceptable CN release enhancement (>50%).

The in vitro lipolysis findings were in good agreement with the corresponding in vitro studies. Liquid SNEDDS showed complete CN release during the experiment beginning and maintained more than 87% CN in solution. This strongly correlates to the enhanced CN release from liquid SNEDDS in the dissolution studies and confirms the robustness of the system under biorelevant conditions. On the other hand, 20% cured S-SNEDDS was able to release ~55–60% of CN in the lipolysis study. This also correlates well with the corresponding dissolution findings that > 50% release from the same system.

The accelerated stability study showed considerable CN degradation within the cured S-SNEDDS formulations. The CN amount in formulation dropped to 70–78% upon 3 months’ storage in accelerated conditions. These findings are in agreement with previous studies that showed significant CN degradation within L-SNEDDS and S- SNEDDS formulations [3,4,5]. These findings were attributed to the direct contact of the drug with the SNEDDS excipients, which led to hydroxylation of CN molecule due to free fatty acid present within the formulation. Solidification into cured S- SNEDDS might have decreased the CN degradation rate but could not stop it completely because the drug was still in direct contact with the S-SNEDDS excipients [34].

Regarding the in vitro dissolution at pH 1.2, the cured S-SNEDDS (20% and 30% PVP) showed about 20% and 11% decrease in drug release at 2 h, upon storage for 3 months (Figure 15A). These data are strongly correlated with the chemical stability data that showed a relative decrease in intact drug amount upon storage for 3 months. In contrast, the in vitro dissolution at pH 6.8 revealed that cured S-SNEDDS (20% and 30%) showed a significant drop in drug release after 3 months. The results analysis showed that cured S-SNEDDS (20% and 30% PVP) showed an ≈ 42% drop in drug release at 2 h (Figure 15B). These findings reveal that the drop in drug release at pH 6.8 was not attributed to chemical degradation only but also to drug and/or SNEDDS impediment within the adsorbent. Previous studies suggested several mechanisms to explain the incomplete release of drug/SNEDDS from the silica adsorbents [8,30].

(i) Patki and Patel reported that certain SNEDDS spontaneously form gel upon contact with the aqueous medium, which hinders complete emulsification and/or drug release from S-SNEDDS due to clogging of the adsorbent pores and impeding the SNEDDS inside [11]. However, this mechanism might not be predominant in the current study because previous studies confirmed that the utilized SNEDD excipients (OL/I308/K-EL) at (25/25/50 % *w*/*w*/*w* ratio) do not tend to form gel upon contact with water [35]. Moreover, if gel formation was the sole reason for the loss in drug release then the incomplete release should have been observed in both freshly prepared and stored samples to the same extent. 

(ii) Patki and Patel also suggested that lipophilic drugs have a high affinity towards the hydrophobic surface of the adsorbent and hence might diffuse from SNEDDS to the adsorbent surface leading to nucleation and drug precipitation [11]. This mechanism might not be the predominant reason for incomplete drug release, in the current study, because CN is a highly lipophilic drug (log *p* = 5.8) that has remarkable solubility in OL (23.7% *w*/*w*) [13,36]. Therefore, it is less likely that CN diffuses from the favorable oil phase of SNEDDS to the adsorbent surface. 

(iii) Gumaste et al. suggested that, in the case of freshly prepared solidified systems, the SNEDDS adsorbed onto the adsorbent were predominantly retained in the macroporous region of the adsorbent, while it gradually traveled deeper into the mesoporous region of the adsorbent upon storage (<50 nm pore size), thus decreasing the level of water uptake and turning the emulsification within the small pores of the adsorbent more difficult [8]. This suggested mechanism is strongly matching with the current study findings as follows. The current BET study showed that SYL showed a gradual decrease in surface area and pore volume upon increasing the PVP ratio from 10 to 30% (Table 3). In particular, the pore volume was significantly reduced (by (≈37%) in cured SYL (30% PVP) compared to uncured SYL, which suggests the successful blocking of the majority of SYL pores by PVP. However, the increase in pore size upon increasing the PVP ratio suggests that the small pores (<50 nm) were not preferentially blocked by PVP. Alternatively, PVP might have predominantly blocked larger pores than small pores, as evidenced by pore size comparison between neat SYL and different curing percentages. These findings are in strong agreement with previous studies that suggested no preferential block of small pores by PVP [8]. In addition, the current stability study was conducted at an elevated temperature (40 °C), which could substantially decrease the SNEDDS viscosity. Gumaste et al. suggested that formulation of lower viscosity could travel deeper into the interior small pores of the adsorbent upon storage. Therefore, SNEDDS could be more difficult to emulsify due to less wettability and decreased room provided for emulsification within the deep small pores [8]. This hypothesis strongly supports the liquisolid theory, which postulates that the liquid system is adsorbed onto SYL particles that apparently look similar to a dry powder, as discussed earlier [37,38]. Finally, the remarkable difference in the loss of drug release in pH 1.2 and 6.8 support this hypothesis. CN is a weak base with a reported higher solubility at low pH (1.2) and very low solubility at neutral pH [13]. Therefore, the CN release at pH 1.2 was less affected by SNEDDS entrapment, within the adsorbent small pores, upon storage, while its dissolution at Ph 6.8 was severely affected by SNEDDS entrapment upon storage as shown in Figure 15.

Accordingly, one or more of the aforementioned mechanisms are thought to cause progressive loss of drug release from cured S-SNEDDS upon storage at accelerated conditions. In fact, the use of silica adsorbent and their curing process for SNEDDS solidification needs more attention as the area is still unexplored substantially. Future studies should involve the use of a lipophilic fluorescent probe to investigate the mechanism of incomplete drug release from freshly prepared as well as stored S-SNEDDS. These studies could help to understand the predominant mechanism of incomplete drug release from freshly prepared and stored S-SNEDDS. In addition, future studies could explore the significance of the physical separation between CN solid dispersion and drug-free cured S-SNEDDS (through capsule-in-capsule technology) on CN release and stability within the formulation [2,39]. Finally, further in vivo pharmacokinetic and pharmacodynamic studies should be conducted to correlate the formulation in vitro findings with the in vivo performance.

It is worth mentioning that the current cured systems minimize the adverse effect of using organic solvents for the adsorbent precoating process. Using such organic solvents in the formulation preparation imparts several health and industrial limitations. Health regulatory authorities require strict tests to confirm that the utilized organic solvents have been properly removed or below the acceptable limits. In contrast, the currently utilized precoating approaches ensured the utilization of aqueous solvents (tagged as green within the scope of the current curing process) all over the formulation development to maintain higher product safety attributes and environmentally benign manufacturing conditions.

## 5. Conclusions

S-SNEDDSs, successfully prepared by the adsorption method, present a valuable approach for enhancing drug dissolution and formulation stability. Nevertheless, the adsorbents used in this technique, such as marketed SYL (uncured), severely impede the release of CN from the formulation. In the current study, SNEDDSs were successfully solidified using various cured SYL adsorbent ratios. The present study provides a novel technique to circumvent the adverse effects of adsorbent on drug release from solidified SNEDDS. Additionally, by enhancing dissolution at elevated pH environments, the developed S-SNEDDS could be a potential dosage form to improve the dissolution of unstable weakly basic drugs in patients with hypochlorhydria. Adsorbent precoating by lyophilization significantly enhanced CN release from the formulation. However, the cured S-SNEDDS showed considerable CN degradation and decrease in drug release upon storage in accelerated conditions.

## Figures and Tables

**Figure 1 pharmaceutics-15-00134-f001:**
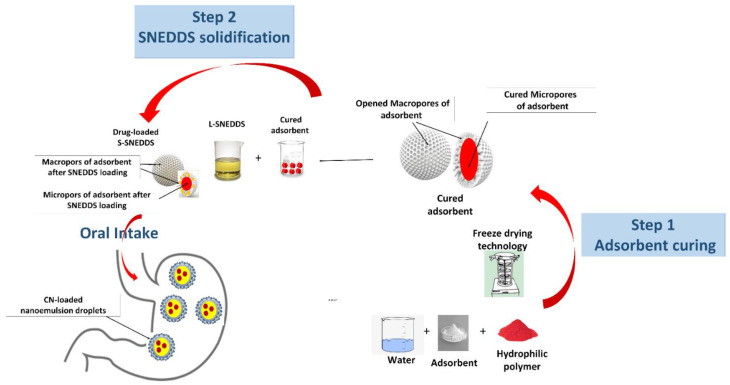
Schematic diagram of the manufacturing process and hypothesized performance of drug-loaded S-SNEDDS (using precoated adsorbent).

**Figure 2 pharmaceutics-15-00134-f002:**
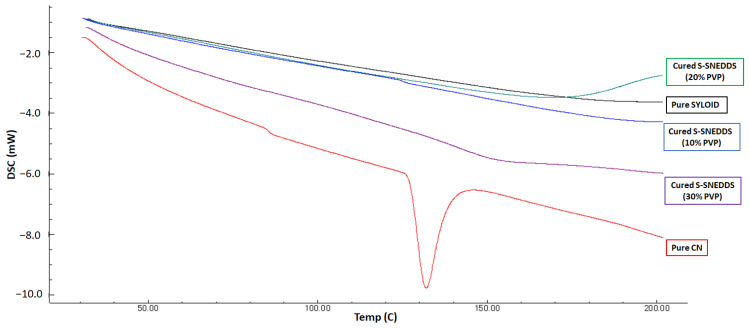
DSC chromatograms of pure CN and cured S-SNEDDS.

**Figure 3 pharmaceutics-15-00134-f003:**
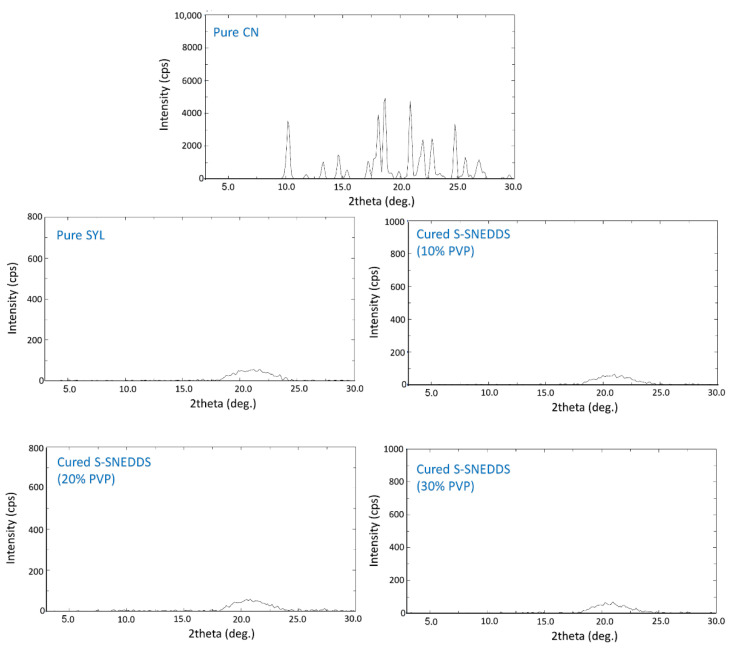
XRD analysis of pure CN and cured S-SNEDDS at different percentages of PVP.

**Figure 4 pharmaceutics-15-00134-f004:**
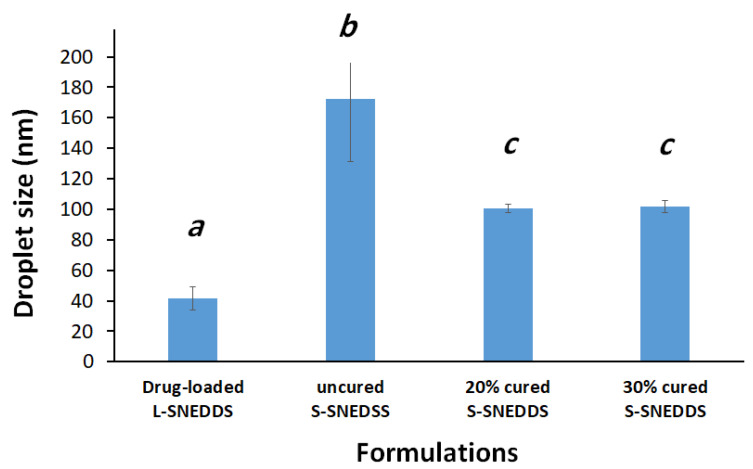
Influence of SNEDDS solidification and adsorbent curing on the formulation droplet size. Different lowercase italic letters (above the bars) indicate significant differences between samples (*p* < 0.05), while samples denoted with a common letter (above the bars) are not significantly different. Data are expressed as mean ± SD.

**Figure 5 pharmaceutics-15-00134-f005:**
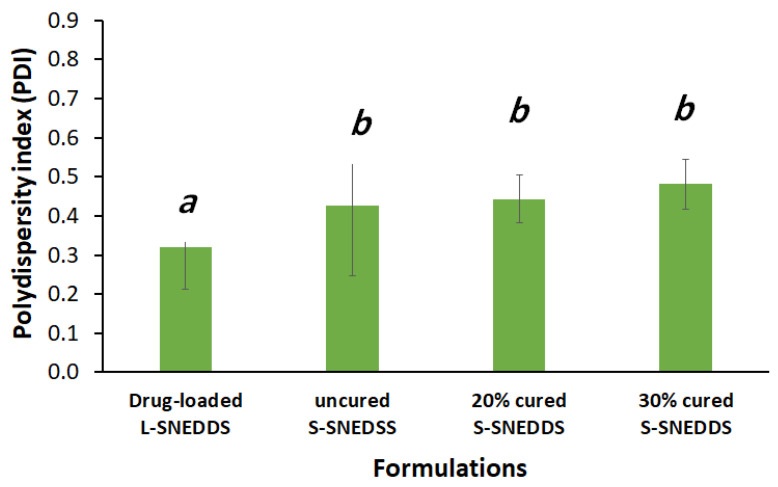
Influence of SNEDDS solidification and adsorbent curing on the polydispersity index (PDI). Different lowercase italic letters (above the bars) indicate significant differences between samples (*p* < 0.05), while samples denoted with a common letter (above the bars) are not significantly different. Data are expressed as mean ± SD.

**Figure 6 pharmaceutics-15-00134-f006:**
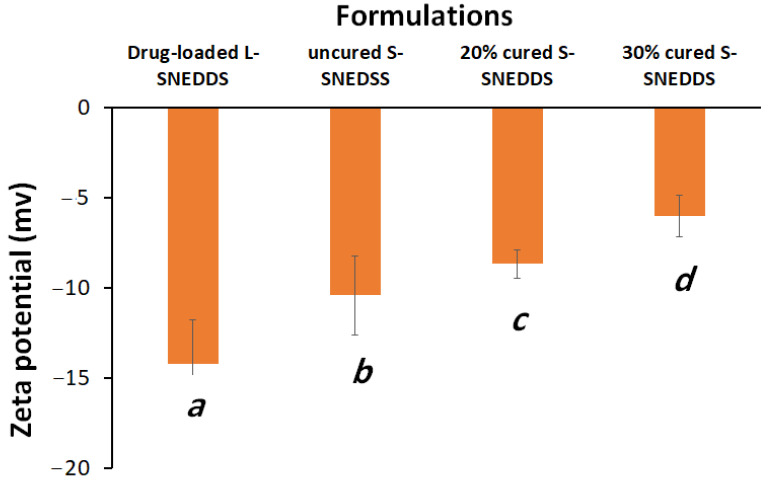
Influence of solidification and curing of solid SNEDDS on formulation zeta potential. Different lowercase italic letters (below the bars) indicate significant differences between samples (*p* < 0.05), while samples denoted with a common letter (below the bars) are not significantly different. Data are expressed as mean ± SD.

**Figure 7 pharmaceutics-15-00134-f007:**
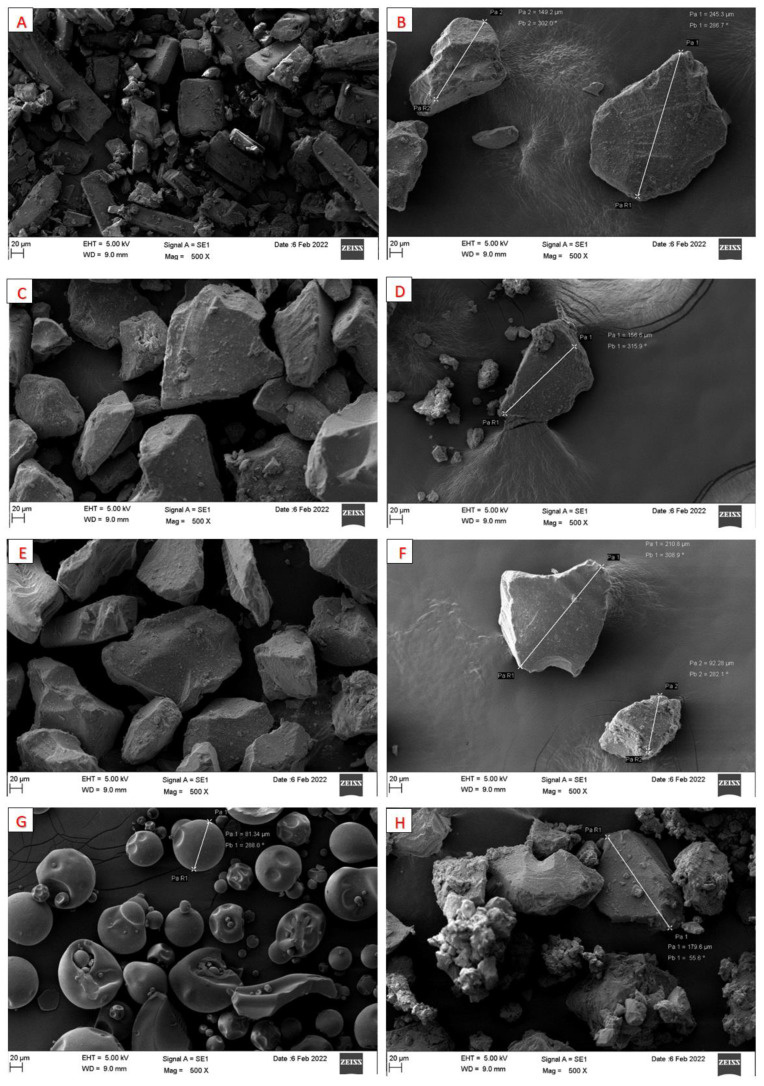
SEM images of (**A**) pure CN, (**B**) pure SYL, (**C**) cured SYL (10% PVP), (**D**) cured S-SNEDDS (10% PVP), (**E**) cured SYL (20% PVP), (**F**) cured S-SNEDDS (20% PVP), (**G**) cured SYL (30% PVP), and (**H**) cured S-SNEDDS (30% PVP) at 500× magnification.

**Figure 8 pharmaceutics-15-00134-f008:**
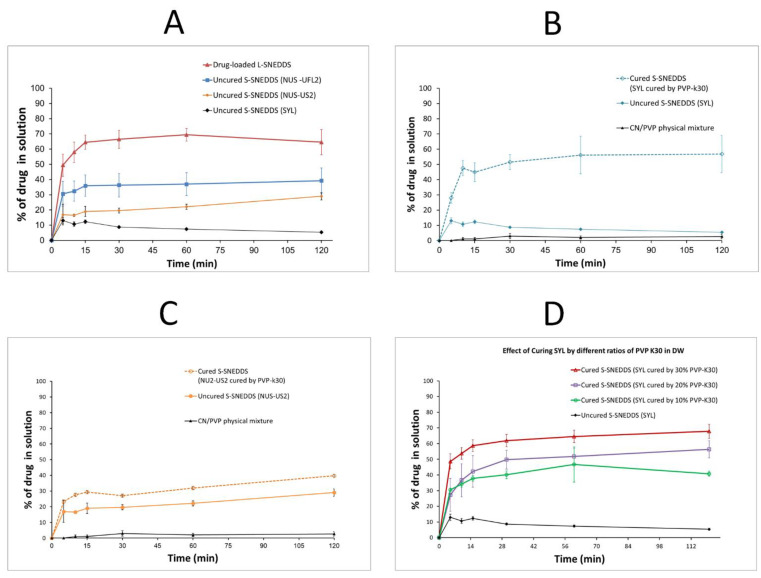
Effect of the (**A**) adsorbent type, (**B**) the adsorbent NUS-US2 curing, (**C**) the adsorbent SYL curing, and (**D**) curing polymer concentration on CN release from S-SNEDDS at pH 6.8. Data are expressed as mean ± SD.

**Figure 9 pharmaceutics-15-00134-f009:**
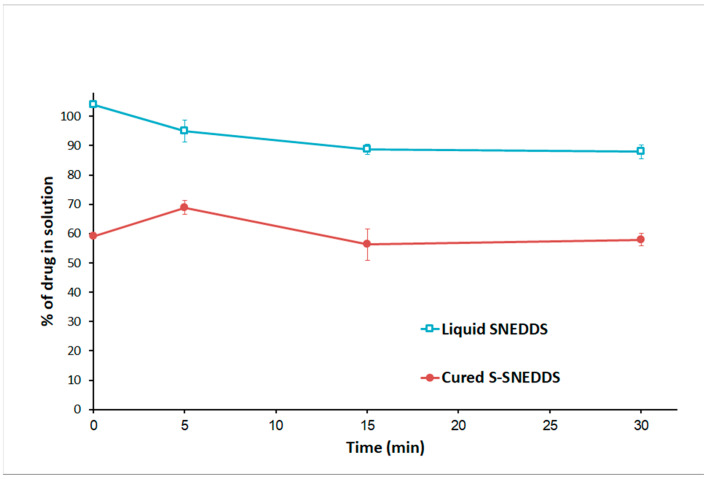
Influence of in vitro lipolysis on the % drug in solution FESSIF conditions. Data are expressed as mean ± SD, *n* = 3. Cured S-SNEDDS was represented by cured S-SNEDDS (comprising SYL cured by 20% PVP).

**Figure 10 pharmaceutics-15-00134-f010:**
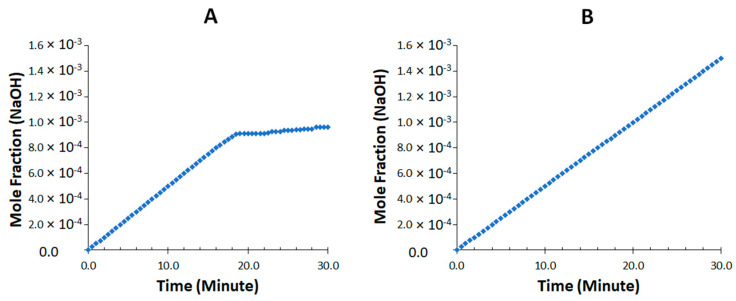
Moles of NaOH (mM) titrated during 30 min digestion period for (**A**) drug-loaded L-SNEDDS and (**B**) cured S-SNEDDS (20% PVP) under fed conditions.

**Figure 11 pharmaceutics-15-00134-f011:**
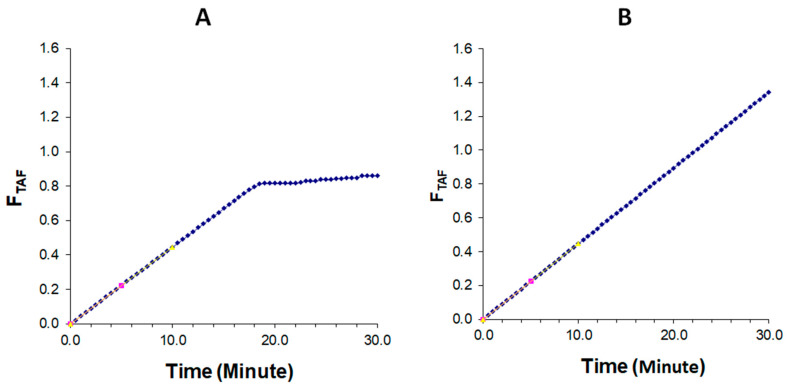
Fraction of total available fatty acid (FA) released during 30 min digestion period for (**A**) drug-loaded L-SNEDDS and (**B**) cured S-SNEDDS (20% PVP) under fed conditions. The yellow dotted arrow shows the initial digestion rate, which was measured as the amount of FA (mmol/min) hydrolyzed during the initial 0–10 min.

**Figure 12 pharmaceutics-15-00134-f012:**
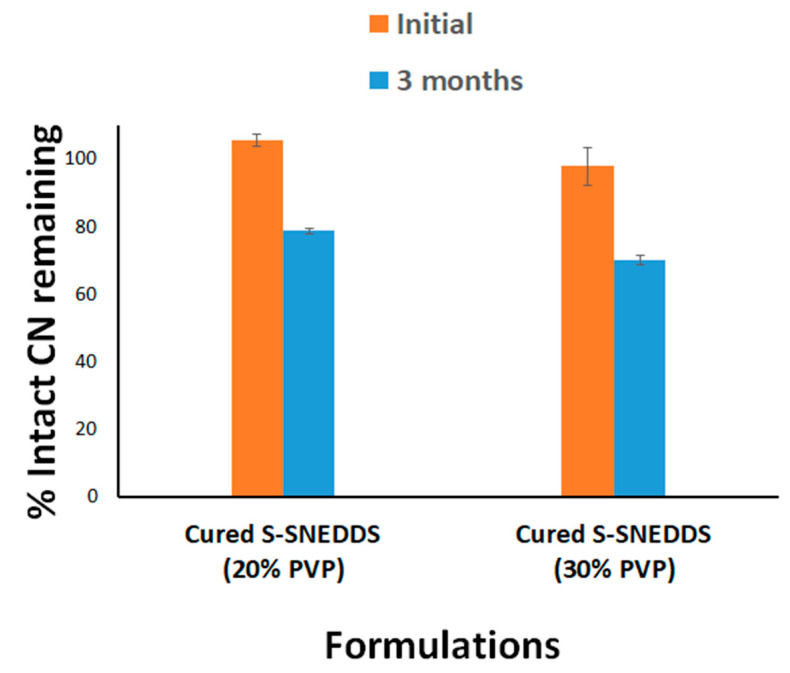
The chemical stability of cured S-SNEDDS at accelerated storage conditions.

**Figure 13 pharmaceutics-15-00134-f013:**
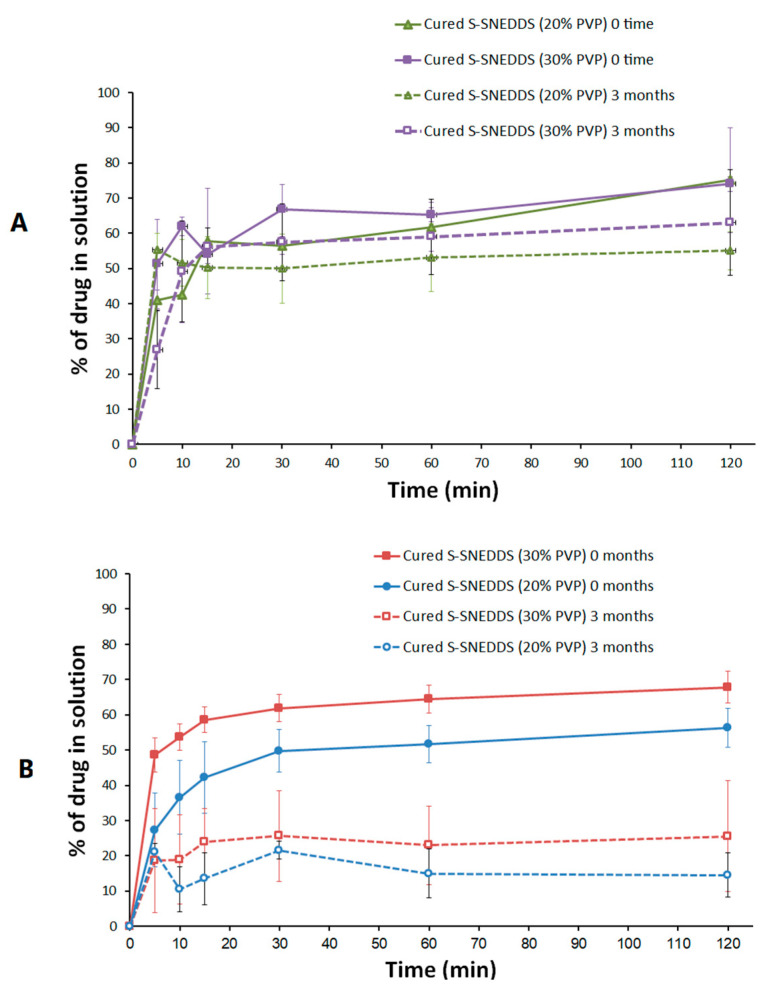
Influence of accelerated storage conditions on the in vitro dissolution of cured S-SNEDDS formulations at (**A**) pH 1.2 and (**B**) pH 6.8. Data are expressed as mean ± SD.

**Figure 14 pharmaceutics-15-00134-f014:**
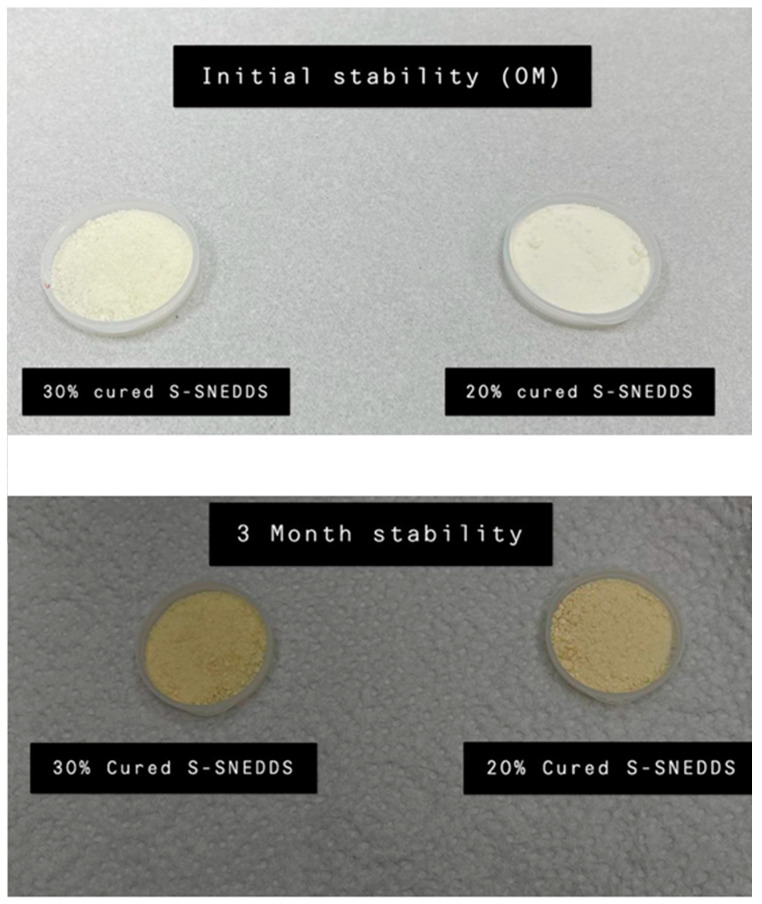
Captured images of cured S-SNEDDS at 0 and 3 months of storage at accelerated conditions.

**Figure 15 pharmaceutics-15-00134-f015:**
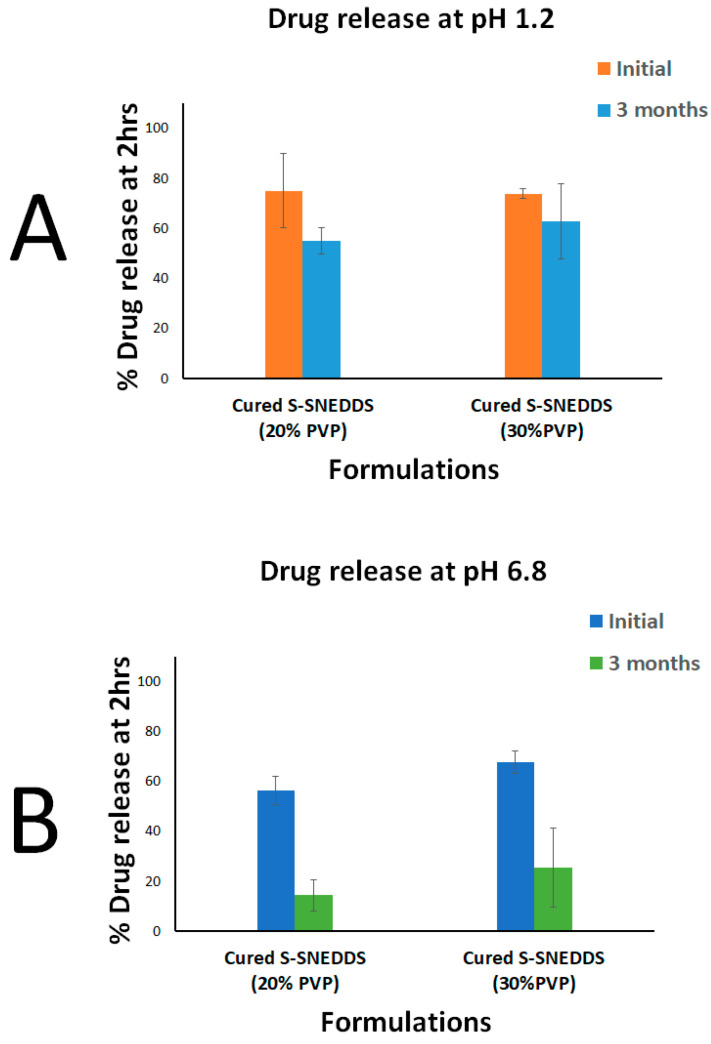
Drug release at (**A**) pH 1.2 and (**B**) pH 6.8. Data are expressed as mean ± SD.

**Table 1 pharmaceutics-15-00134-t001:** Various liquid and solid formulations for CN using lipid-based excipients and inorganic silica adsorbents.

Excipients *	Formulations				
	Drug-Loaded L- SNEDDS	Uncured S-SNEDDS	10% Cured S-SNEDDS	20% Cured S-SNEDDS	30% Cured S-SNEDDS
CN	8	4	4	4	4
Oleic acid	23	11.5	11.5	11.5	11.5
Imwitor I308	23	11.5	11.5	11.5	11.5
Kolliphor El	46	23	23	23	23
Syloid	-	50	45	40	35
PVP-K30	-	-	5	10	15
SUM	100	100	100	100	100

* All the excipients’ quantities are expressed as *w*/*w* %. Abbreviations: L-SNEDDS and S-SNEDDS: liquid and solid self-nanoemulsifying drug delivery systems, respectively.

**Table 2 pharmaceutics-15-00134-t002:** Evaluation of powder flow properties of different ratios of cured adsorbent before and after SNEDDS addition.

Formulation	Test Attributes	Angle of Repose	Bulk Density	Tapped Density	Compressibility Index	Hausner Ratio
Pure SYL	Value	37.2 ± 0.1	0.27 ± 0.01	0.23 ± 0.01	9.09 ± 0.37	1.10 ± 0.01
	Flow property *	Fair	-	-	Excellent	Excellent
Cured SYL (10%PVP)	Value	41 ± 0.1	0.19 ± 0.01	0.23 ± 0.01	18.75 ± 0.37	1.23 ± 0.01
	Flow property *	passable	-	-	Fair	Fair
Cured SYL (20%PVP)	Value	41.6 ± 0.1	0.19 ± 0.01	0.25 ± 0.01	18.75 ± 0.37	1.33 ± 0.01
	Flow property *	passable	-	-	passable	passable
Cured SYL (30%PVP)	Value	46.6 ± 0.1	0.20 ± 0.01	0.27 ± 0.01	26.67 ± 0.37	1.36 ± 0.01
	Flow property *	poor	-	-	poor	poor
Cured S-SNEDDS (10% PVP)	Value	39.9 ± 0.1	0.38 ± 0.01	0.43 ± 0.01	12.50 ± 0.37	1.14 ± 0.01
	Flow property *	Fair	-	-	Good	Good
Cured S-SNEDDS (20% PVP)	Value	43.3 ± 0.1	0.38 ± 0.01	0.43 ± 0.01	12.50 ± 0.37	1.14 ± 0.01
	Flow property *	passable	-	-	Good	Good
Cured S-SNEDDS (30% PVP)	Value	44.5 ± 0.1	0.27 ± 0.01	0.33 ± 0.01	18.18 ± 0.37	1.22 ± 0.01
	Flow property *	passable	-	-	Fair	Fair

* Categorized according to USP 35, Powder flow chapter [28]. Data are expressed as mean ± SD, *n* = 3.

**Table 3 pharmaceutics-15-00134-t003:** The Brunauer–Emmett–Teller of different formulas.

Sample	Total Surface Area (m²/g) *	Pore Volume (cm^3^/g) **	Average Pore Size (Å) ***
Pure SYL (uncured)	309.5	1.83	239.5
Cured SYL (10% PVP)	271.4	1.51	226.3
Cured SYL (20% PVP)	266.6	1.44	219.5
Cured SYL (30% PVP)	236.9	1.15	205.3

* Total surface area was calculated based on the BET surface area or t-Plot external surface area. ** Pore volume was calculated based on BJH. Adsorption cumulative volume of pores between 17,000 Å and 3,000,000 Å diameter: *** Average pore size was calculated as adsorption average pore width (4 V/A by BET).

## Data Availability

The data presented in this study are available within the article.

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
