# Peer review of "Adsorbent Precoating by Lyophilization: A Novel Green Solvent Technique to Enhance Cinnarizine Release from Solid Self-Nanoemulsifying Drug Delivery Systems (S-SNEDDS)"

_pharmaceutics, 2022, doi:10.3390/pharmaceutics15010134_

Round 1
Reviewer 1 Report
The authors investigate silica with large surface area to adsorb a drug on it, later coat it with a film (emulsion) and release the drug in simulated stomach fluid. The work is very important for the society and requires only minor revision prior to publication.
1. SNEDDS should be stated fully also in Abstract not just header
2. Please use spacing between words.
3. Some points (sentence separators) were omitted.
4. Figure 4: Please define a – c in Figure caption. Why are there so many c?
5. Figure 4: Please define a and b in Figure caption. Why are there so many b?
6. Figure 4: Please define a and d in Figure caption.
7. Judging the size of the drug delivery systems in Figure 7 G it is fair to compare the particles with other systems of similar size like those for drug delivery to the stomach [1], the authors should mention differences to this competitive system. Other authors work with microsystems [2] which is intermediate to true nano. The authors could mention differences to this one as well for comparison.
8. Figure 10: A and B are of different size.
9. Figure 11: A and B are of different size.
10. The authors should give an outlook. Do they plan cell and / or animal studies??
References
[1] S. Rutkowski, T. Si, M. Gai, M. Sun, J. Frueh, Q. He, Magnetically-guided Hydrogel Capsule Motors produced via Ultrasound assisted Hydrodynamic Electrospray Ionization Jetting, J. Colloid Interface Sci. 25 (2019) 752–774. https://doi.org/10.1016/j.jcis.2019.01.103.
[2] G.B. Sukhorukov, E. Donath, H. Lichtenfeld, E. Knippel, M. Knippel, A. Budde, H. Mohwald, Layer-by-layer self assembly of polyelectrolytes on colloidal particles, Colloids Surfaces A Physicochem. Eng. Asp. 137 (1998) 253–266. https://doi.org/10.1016/S0927-7757(98)00213-1.
Author Response
Response to the Reviewer’s Comments-1
The comments of Editor and reviewers were valuable and helped to improve the quality of the manuscript. Due to the addition of a new figure and references, the reviewers are kindly requested to track the movement of the figures/sections/references. Below is a summary where all the comments have been addressed point-by-point, as requested by the Editor-in-Chief:
Reviewer`s 1 comments
The authors investigate silica with large surface area to adsorb a drug on it, later coat it with a film (emulsion) and release the drug in simulated stomach fluid. The work is very important for the society and requires only minor revision prior to publication.
1. SNEDDS should be stated fully also in abstract not just header
Response 1: According to the reviewer`s comment, the full term "self-nanoemulsifying drug delivery systems" was stated before the abbreviation SNEDDS.
2. Please use spacing between words.
Response 2: The manuscript was thoroughly revised, spacing between words was added and typing mistakes were corrected.
3. Some points (sentence separators) were omitted.
Response 3. The manuscript was thoroughly revised, sentence separators were added and typing mistakes were corrected.
4. Figure 4: Please define a – c in Figure caption. Why are there so many c?
Response 4: the a-c letters utilized in Figure 4 aim to highlight whether values are statistically different from each other or not. This method to highlight significance was adapted from a previous publication that discussed this issue thoroughly [1]. In Figure 4, different lowercase italic letters (above the bars) indicate significant differences between samples (p < 0.05) while samples denoted with a common letter (above the bars) are not significantly different (Figure 4 caption). In particular, 20% cured S-SNEDDS and 30% cured S-SNEDDS are not significantly different so they share the common letter "C" while both of them are statistically different from drug-loaded L-SNEDDS and uncured S-SNEDDS which were denoted by different letters "a" and "b", respectively.
5. Figure 5: Please define a and b in Figure caption. Why are there so many b?
Response 5: the a-b letters utilized in Figure 5 aim to highlight whether values are statistically different from each other or not. This method to highlight significance was adapted from a previous publication that discussed this issue thoroughly [1]. In Figure 5, different lowercase italic letters (above the bars) indicate significant differences between samples (p < 0.05) while samples denoted with a common letter (above the bars) are not significantly different (Figure 5 caption). In particular, uncured S-SNEDDS, 20% cured S-SNEDDS, and 30% cured S-SNEDDS are not significantly different so they share the common letter "b" while all of them are statistically different from drug-loaded L-SNEDDS which were denoted a different letter "a".
6. Figure 6: Please define a and d in Figure caption.
Response 6: the a-d letters utilized in Figure 6 aim to highlight whether values are statistically different from each other or not. This method to highlight significance was adapted from a previous publication that discussed this issue thoroughly [1]. In Figure 6, different lowercase italic letters (above the bars) indicate significant differences between samples (p < 0.05) while samples denoted with a common letter (above the bars) are not significantly different (Figure 6 caption). In particular, drug-loaded L-SNEDDS, uncured S-SNEDDS, 20% cured S-SNEDDS and 30% cured S-SNEDDS are all statistically different from each other so they were denoted by different letters "a", "b", "c" and "d", respectively.
7. Judging the size of the drug delivery systems in Figure 7 G it is fair to compare the particles with other systems of similar size like those for drug delivery to the stomach [2], the authors should mention differences to this competitive system. Other authors work with microsystems [3] which is intermediate to true nano. The authors could mention differences to this one as well for comparison.
Response 7. According to the reviewer`s comment, the following paragraph was added in the "Discussion" section.
The SEM data (Figure 7) reveals that the adsorbent SYL and cured S-SNEDDS showed discrete irregular particles that ranged in size from a few micrometers up to ≈245 µm. Upon exposure to GIT aqueous media, these microparticles are expected to undergo a rapid self-nanoemulsification process resulting in a fine nanoemulsion with an average size of ≈101 nm, as evidenced by the formulation droplet size findings (Figure 4). This in-situ transition from the microscale to nanoscale is valuable in terms of dissolution and absorption prospects. This ultra-low nano droplet size could be linked with increased drug release rate, increased surface area available for drug absorption, and formation of readily digestible oil droplets that can be incorporated into mixed micelles and pass the intestinal lumen [4,5]. Compared to other systems of similar size, Rutkowski et al reported a technique to prepare hydrogel alginate capsules (microparticles) that can be adjusted in size from 10 µm to 2 mm [2]. In addition, Sukhorokov reported a method to prepare multi-layer films of polyelectrolytes adsorbed onto charged polystyrene latex particles. These nanoparticles ranged in size from 100-260 nm depending on the number of layers applied [3]. In fact, the aforementioned microparticles [2] and nanoparticles [3] are quite different drug delivery systems that have different pharmaceutical applications other than the currently studied S-SNEDDS. However, it is worth mentioning that these cured S-SNEDDS offer an exclusive advantage of in-situ transition from the micro to nanoscale upon exposure to GIT aqueous environment. Accordingly, the current cured S-SNEDDS gather the advantage of microparticles in terms of better physical stability and less particle aggregation on storage along with the advantages of nanoparticles (that are formed in-situ upon the self-emulsification process) in terms of significantly higher surface area available for drug absorption.
8. Figure 10: A and B are of different size.
Response 8. Figure 10 A and B were modified to be in the same size
9. Figure 11: A and B are of different size.
Response 9. Figure 11 A and B were modified to be in the same size
10. The authors should give an outlook. Do they plan cell and / or animal studies??
Response 10. According to the reviewer`s request, the following statement was added at the end of the "Discussion" section.
Finally, further in-vivo pharmacokinetic and pharmacodynamic studies should be conducted to correlate the formulation in-vitro findings with the in-vivo performance.
References
-
Piepho, H.P. Letters in Mean Comparisons: What They Do and Don’t Mean. Agron. J. 2018, 110, 431–434, doi:10.2134/AGRONJ2017.10.0580.
-
Rutkowski, S.; Si, T.; Gai, M.; Sun, M.; Frueh, J.; He, Q. Magnetically-guided hydrogel capsule motors produced via ultrasound assisted hydrodynamic electrospray ionization jetting. J. Colloid Interface Sci. 2019, 541, 407–417, doi:10.1016/J.JCIS.2019.01.103.
-
Sukhorukov, G.B.; Donath, E.; Lichtenfeld, H.; Knippel, E.; Knippel, M.; Budde, A.; Möhwald, H. Layer-by-layer self assembly of polyelectrolytes on colloidal particles. Colloids Surfaces A Physicochem. Eng. Asp. 1998, 137, 253–266, doi:10.1016/S0927-7757(98)00213-1.
-
Baloch, J.; Sohail, M.F.; Sarwar, H.S.; Kiani, M.H.; Khan, G.M.; Jahan, S.; Rafay, M.; Chaudhry, M.T.; Yasinzai, M.; Shahnaz, G. Self-Nanoemulsifying Drug Delivery System (SNEDDS) for Improved Oral Bioavailability of Chlorpromazine: In Vitro and In Vivo Evaluation. Medicina (Kaunas). 2019, 55, doi:10.3390/medicina55050210.
-
Kazi, M.; Shahba, A.A.; Alrashoud, S.; Alwadei, M.; Sherif, A.Y.; Alanazi, F.K. Bioactive self-nanoemulsifying drug delivery systems (Bio-SNEDDS) for combined oral delivery of curcumin and piperine. Molecules 2020, 25, 1–26, doi:10.3390/molecules25071703.
Reviewer 2 Report
The manuscript contains a technique to enhance the release of cinnarizine from a solid self- nanoemulsifying drug delivery system (S-SNEDD) using adsorbent pre-coating by freeze-drying. Thus, it is suitable topic for publication in the journal "Pharmaceutics ". However, it is thought to be accepted with major revision because this manuscript has the following problems.
1. The solid SEDDS system study using cinnarizine has been studied for a long time, and in this manuscript, solidification and increased dissolution are shown as the main parts through pre-coating of the adsorbent by PVP by freeze drying. This paper is an interesting subject. However, the use of PVP is an excipient that has been used for a long time to increase dissolution. Therefore, a simple mixture of PVP is expected to have a similar effect. It is considered important to confirm the difference between uncured S-SNEEDS and a physical mixture of PVP.
2. Decreased dissolution after stability evaluation is considered a major weakness of this study. Further discussion is required.
3. It is essential to prove the increased dissolution in vitro by evaluating the bioavailability in vivo.
The manuscript contains a technique to enhance the release of cinnarizine from a solid self- nanoemulsifying drug delivery system (S-SNEDD) using adsorbent pre-coating by freeze-drying. Thus, it is suitable topic for publication in the journal "Pharmaceutics ". However, it is thought to be accepted with major revision because this manuscript has the following problems.
1. The solid SEDDS system study using cinnarizine has been studied for a long time, and in this manuscript, solidification and increased dissolution are shown as the main parts through pre-coating of the adsorbent by PVP by freeze drying. This paper is an interesting subject. However, the use of PVP is an excipient that has been used for a long time to increase dissolution. Therefore, a simple mixture of PVP is expected to have a similar effect. It is considered important to confirm the difference between uncured S-SNEEDS and a physical mixture of PVP.
2. Decreased dissolution after stability evaluation is considered a major weakness of this study. Further discussion is required.
3. It is essential to prove the increased dissolution in vitro by evaluating the bioavailability in vivo.

Author Response
Response to the Editor and Reviewer’s Comments-2
The comments of Editor and reviewers were valuable and helped to improve the quality of the manuscript. Due to the addition of a new figure and references, the reviewers are kindly requested to track the movement of the figures/sections/references. Below is a summary where all the comments have been addressed point-by-point, as requested by the Editor-in-Chief:
Reviewer 2 comments
The manuscript contains a technique to enhance the release of cinnarizine from a solid self- nanoemulsifying drug delivery system (S-SNEDD) using adsorbent pre-coating by freeze-drying. Thus, it is suitable topic for publication in the journal "Pharmaceutics ". However, it is thought to be accepted with major revision because this manuscript has the following problems.
- The solid SEDDS system study using cinnarizine has been studied for a long time, and in this manuscript, solidification and increased dissolution are shown as the main parts through pre-coating of the adsorbent by PVP by freeze drying. This paper is an interesting subject. However, the use of PVP is an excipient that has been used for a long time to increase dissolution. Therefore, a simple mixture of PVP is expected to have a similar effect. It is considered important to confirm the difference between uncured S-SNEEDS and a physical mixture of PVP.
Response 1. According to the reviewer`s request, a physical mixture of PVP/CN was prepared and evaluated in terms of in-vitro dissolution compared to uncured and cured S-SNEDDS. The following paragraphs were added in the manuscript to describe these findings in the" methods", "results" and "discussion" sections as follow:
In the current dissolution study, CN/PVP-K30 physical mixture (≈1:2.5 w/w) was utilized as a control, and various uncured and cured SNEDDS (≈25 mg CN equivalent) were examined to evaluate the influence of different curing parameters on CN release (Methods section).
PVP physical mixture showed negligible CN release (up 3%) within 2hr. Both uncured S-SNEDDS (comprising SYL and NUS-US2) showed superior CN release (up to 13% and 29%, respectively) compared to PVP physical mixture (Figure 8B, C) ("Results" section).
Regarding the in-vitro dissolution studies, the CN/PVP physical mixture showed negligible CN release at pH 6.8. Although PVP-K30 is an effective precipitation inhibitor that has been widely used to enhance drug dissolution, it failed to enhance CN dissolution (at ph 6.8), when used solely. This finding might be owing to the challenging physicochemical property of CN; being a weak base with very poor solubility at neutral and basic media. Similar findings were reported in previous studies where CN/PVP solid dispersion showed significant precipitation and limited dissolution at pH 6.8 [1]. However, SNEDDS technology showed significant enhancement of CN dissolution at the same pH. This might be attributed to SNEDDS ability to form a favorable microenvironment within nanoemulsions droplets that were able to maintain CN solubilized and protect it from exposure to the unfavorable neutral environment that is associated with dramatic drug precipitation and limited release ("Discssion" section).
In contrast to CN/PVP physical mixture, the proper utilization of PVP-K30 in adsorbent pre-coating led to significant CN release enhancement. This finding confirms the vital role of PVP in the adsorbent pre-coating which helped in overcoming the adsorbent adverse effect on drug release from solidified SNEDDS. On the other hand, the sole presence of PVP in the formula failed to enhance CN release while PVP played a vital role in overcoming the significant retardation of drug release from SNEDDS that has been attributed to the adsorbent ("Discussion" section).
- Decreased dissolution after stability evaluation is considered a major weakness of this study. Further discussion is required.
Response 2. According to the reviewer`s comment, a comprehensive discussion was included to explore the possible mechanisms of decreased drug release upon storage. In addition, the pore volume values and a new figure were added to strengthen the discussion of these findings (Table 3, Figure 15). Finally, A future outlook of the next potential studies was added in the end of the discussion to open a window for further investigation of the current technique.
Accordingly, the following paragraphs were added in the "Discussion" section:
Regarding the in-vitro dissolution at pH 1,2, the cured S-SNEDDS (20% and 30% PVP) showed about 20% and 11% decrease in drug release at 2hr, upon storage for 3 months (Figure 15A). These data are strongly correlated with the chemical stability data that showed a relative decrease in intact drug amount upon storage for 3 months. In contrast, the in-vitro dissolution at pH 6.8 revealed that cured S-SNEDDS (20% and 30%) showed a significant drop in drug release after 3 months. The results analysis showed that cured S-SNEDDS (20% and 30% PVP) showed ≈ 42% drop in drug release at 2hr (Figure 15B). These findings reveal that the drop in drug release at pH 6.8 was not attributed to chemical degradation only but also to drug and/or SNEDDS impediment within the adsorbent. Previous studies suggested several mechanisms to explain the incomplete release of drug/SNEDDS from the silica adsorbents [2,3].
(i) Patki and Patel reported that certain SEDDS spontaneously form gel upon contact with the aqueous medium which hinders complete emulsification and/or drug release fromS-SNEDDS due to clogging of the adsorbent pores and impeding the SNEDDS inside [4]. However, this mechanism might not be predominant in the current study because previous studies confirmed that the utilized SNEDD excipients (OL/I308/K-EL) at (25/25/50 w/w ratio) do not tend to form gel upon contact with water [5]. Moreover, if gel formation was the sole reason for the loss in drug release then the incomplete release should have been observed in both freshly prepared and stored samples to the same extent.
(ii) Patki and Patel also suggested that lipophilic drugs have a high affinity towards the hydrophobic surface of the adsorbent and hence might diffuse from SNEDDS to the adsorbent surface leading to nucleation and drug precipitation. This mechanism might not be the predominant reason for incomplete drug release, in the current study, because CN is a highly lipophilic drug (log p = 5.8) that has remarkable solubility in OL (23.7% w/w) [6,7]. Therefore, it is less likely that CN diffuses from the favorable oil phase of SNEDDS to the adsorbent surface.
(iii) Gumaste et al suggested that, in the case of freshly prepared solidified systems, the SNEDDS adsorbed onto the adsorbent were predominantly retained in the macroporous region of the adsorbent. While it gradually traveled deeper into the mesoporous region of the adsorbent upon storage (< 50 nm pore size), thus decreasing the level of water uptake and turning the emulsification within the small pores of the adsorbent more difficult. This suggested mechanism is strongly matching with the current study findings as follows. The current BET study showed that SYL showed a gradual decrease in surface area and pore volume upon increasing the PVP ratio from 10 to 30% (Table 3). In particular, the pore volume was significantly reduced (by (≈37%) in cured SYL (30% PVP) compared to uncured SYL which suggests the successful blocking of the majority of SYL pores by PVP. However, the increase in pore size upon increasing the PVP ratio suggests that the small pores (< 50 nm) were not preferentially blocked by PVP. Alternatively, PVP might have predominantly blocked larger pores than small pores as evidenced by pore size comparison between neat SYL and different curing percentages. These findings are in strong agreement with previous studies that suggested no preferential block of small pores by PVP [2]. In addition, the current stability study was conducted at an elevated temperature (40°C) which could substantially decrease the SNEDDS viscosity. Gumaste et al suggested that formulation of lower viscosity could travel deeper into the interior small pores of the adsorbent upon storage. Therefore, SNEDDS could be more difficult to emulsify due to less wettability and decreased room provided for emulsification within the deep small pores [2]. This hypothesis strongly supports the liquisolid theory which postulates that the liquid system is adsorbed onto SYL particles that apparently look like a dry powder, as discussed earlier [8,9]. Finally, the remarkable difference in the loss of drug release in pH 1.2 and 6.8 support this hypothesis. CN is a weak base with a reported higher solubility at low pH (1.2) and very low solubility at neutral pH [6]. Therefore, CN release at pH 1.2 was less affected by SNEDDS entrapment, within the adsorbent small pores, upon storage. While, its dissolution at Ph 6.8 was severely affected by SNEDDS entrapment upon storage as shown in Figure 15.
Accordingly, one or more of the aforementioned mechanisms are thought to cause progressive loss of drug release from cured S-SNEDDS upon storage at accelerated conditions. In fact, the use of silica adsorbent and their curing process for SNEDDS solidification needs more attention as the area is still unexplored substantially. Future studies should involve the use of a lipophilic fluorescent probe to investigate the mechanism of incomplete drug release from freshly prepared as well as stored S-SNEDDS. These studies could help to understand the predominant mechanism of incomplete drug release from freshly prepared and stored S-SNEDDS. In addition, future studies could explore the significance of the physical separation between CN solid dispersion and drug-free cured S-SNEDDS (through capsule-in-capsule technology) on CN release and stability within the formulation [10,11].
- It is essential to prove the increased dissolution in vitro by evaluating the bioavailability in vivo.
Response 3. The reviewer`s comment is highly appreciated and indeed in-vivo animal and/or human studies are required to achieve a solid conclusion about the significance of any pharmaceutical technique in drug formulation. However, within the scope of the studies, this article has not covered the in vivo studies but instead has focused on in vitro investigations of novel techniques to solve a serious drawback of adsorbent utilization in SNEDDS solidification. According to the authors point of view, the use of silica adsorbent and their curing process for SNEDDS solidification needs more attention as the area is still unexplored substantially. Comprehensive in-vitro mechanistic studies could be currently more valuable to deeply investigate the real mechanisms for incomplete drug release from these systems. Due to the article length limitations, the authors preferred not to present other study experiments and findings to avoid possible reader`s confusion with more data. Keeping in mind the amount of time, experimental work, analytical method development for plasma samples, and ethical approval required to conduct animal studies, it was difficult to conduct the in-vivo study within the time usually allowed for manuscript revision. Accordingly, in-vitro lipolysis studies were added in order to simulate gastrointestinal conditions. It is worth mentioning also that numerous valuable and highly cited articles have solely presented the in-vitro performance of solidified SNEDDS with no in-vivo investigation such as [2,4,8,12,13].
Finally, the following statement was included to give an outlook of future in-vivo studies:
Finally, further in-vivo pharmacokinetic and pharmacodynamic studies should be conducted to correlate the formulation in-vitro findings with the in-vivo performance.
References
- Shahba, A.A.W.; Ahmed, A.R.; Alanazi, F.K.; Mohsin, K.; Abdel-Rahman, S.I. Multi-Layer Self-Nanoemulsifying Pellets: an Innovative Drug Delivery System for the Poorly Water-Soluble Drug Cinnarizine. AAPS PharmSciTech 2018, 19, 2087–2102, doi:10.1208/s12249-018-0990-7.
- Gumaste, S.G.; Freire, B.O.S.; Serajuddin, A.T.M. Development of solid SEDDS, VI: Effect of precoating of Neusilin® US2 with PVP on drug release from adsorbed self-emulsifying lipid-based formulations. Eur. J. Pharm. Sci. 2017, 110, 124–133, doi:10.1016/j.ejps.2017.02.022.
- Gumaste, S.G.; Serajuddin, A.T.M. Development of solid SEDDS, VII: Effect of pore size of silica on drug release from adsorbed self-emulsifying lipid-based formulations. Eur. J. Pharm. Sci. 2017, doi:10.1016/j.ejps.2017.05.014.
- Patki, M.; Patel, K. Development of a solid supersaturated self-nanoemulsifying preconcentrate (S-superSNEP) of fenofibrate using dimethylacetamide and a novel co-processed excipient. Drug Dev. Ind. Pharm. 2019, 45, 405–414, doi:10.1080/03639045.2018.1546311.
- Shahba, A.A.; Mohsin, K.; Alanazi, F.K. The studies of phase equilibria and efficiency assessment for self-emulsifying lipid-based formulations. AAPS PharmSciTech 2012, 13, 522–533, doi:10.1208/s12249-012-9773-8.
- Shahba, A.A.W.; Mohsin, K.; Alanazi, F.K. Novel self-nanoemulsifying drug delivery systems (SNEDDS) for oral delivery of cinnarizine: Design, optimization, and in-vitro assessment. AAPS PharmSciTech 2012, 13, 967–977, doi:10.1208/s12249-012-9821-4.
- Tokumura, T.; Tsushima, Y.; Tatsuishi, K.; Kayano, M.; Machida, Y.; Nagai, T. Enhancement of the oral bioavailability of cinnarizine in oleic acid in beagle dogs. J Pharm Sci 1987, 76, 286–288.
- Dalal, L.; Allaf, A.W.; El-Zein, H. Formulation and in vitro evaluation of self-nanoemulsifying liquisolid tablets of furosemide. Sci. Reports 2021 111 2021, 11, 1–10, doi:10.1038/s41598-020-79940-5.
- Hentzschel, C.M.; Sakmann, A.; Leopold, C.S. Suitability of various excipients as carrier and coating materials for liquisolid compacts. http://dx.doi.org/10.3109/03639045.2011.564184 2011, 37, 1200–1207, doi:10.3109/03639045.2011.564184.
- Shahba, A.A.; Tashish, A.Y.; Alanazi, F.K.; Kazi, M. Combined self-nanoemulsifying and solid dispersion systems showed enhanced cinnarizine release in hypochlorhydria/achlorhydria dissolution model. Pharmaceutics 2021, 13, 1–19, doi:10.3390/pharmaceutics13050627.
- Shahba, A.A.W.; Sherif, A.Y.; Elzayat, E.M.; Kazi, M. Combined Ramipril and Black Seed Oil Dosage Forms Using Bioactive Self-Nanoemulsifying Drug Delivery Systems (BIO-SNEDDSs). Pharmaceuticals 2022, 15, 1–2, doi:10.3390/ph15091120.
- Gumaste, S.G.; Dalrymple, D.M.; Serajuddin, A.T.M. Development of solid SEDDS, V: Compaction and drug release properties of tablets prepared by adsorbing lipid-based formulations onto neusilin® US2. Pharm. Res. 2013, 30, 3186–3199, doi:10.1007/s11095-013-1106-4.
- Gumaste, S.G.; Serajuddin, A.T.M. Development of solid SEDDS, VII: Effect of pore size of silica on drug release from adsorbed self-emulsifying lipid-based formulations. Eur. J. Pharm. Sci. 2017, 110, 134–147, doi:10.1016/j.ejps.2017.05.014.
Editor comments:
- We noticed that the figures in the manuscript are with low resolution, please provide the sharper version of all figures.
Response 1. All the figures were revised and higher resolution versions of all the figures were added in the manscript.
- We noticed that Ahmad Yousef Tashish's e-mail is not an institutional e-mail. Please understand we generally do not accept @foxmail,@yahoo,@sohu, @sina, @hotmail email address as sometimes our emails are blocked by these domains. Institutional email, @gmail, @126 and @163 is fine with us. Please provide new email of "Ahmad Yousef Tashish".
Respose 2. Ahmad Tashish`s institutional email was provided in the manuscript.
- We noticed that your manuscript has a little high-duplication at the step of duplication checking. We understand that sometimes a similar expression is inevitable, but try to rephrase those sentences. For your reference, we havehighlighted the texts with higher duplication in the attached file. Please try your best to rewrite those parts to make them are literally different.
Response 3. According to the Editor`s comment, all the slightly duplicated parts were revised and paraphrased (see highlighted paragraphs).
Round 2
Reviewer 2 Report
The authors have shown a lot of efforts to improve the manuscript and this should be well appreciated. I found the authors have addressed all my comments carefully and in detail. As a result, I now recommend the current form can be accepted for publication without further modification.